# Polarization Properties of Coherently Superposed Rayleigh Backscattered Light in Single-Mode Fibers

**DOI:** 10.3390/s23187769

**Published:** 2023-09-08

**Authors:** Hui Dong, Hailiang Zhang, Dora Juan Juan Hu

**Affiliations:** Institute for Infocomm Research (I^2^R), Agency for Science, Technology and Research (A*STAR), 1 Fusionopolis Way, #21-01, Connexis South Tower, Singapore 138632, Singapore; zhang_hailiang@i2r.a-star.edu.sg (H.Z.); jjhu@i2r.a-star.edu.sg (D.J.J.H.)

**Keywords:** optical polarization, fiber optics, Rayleigh scattering, phase-sensitive optical time-domain reflectometry, Jones matrix, Mueller matrix, state of polarization, degree of polarization

## Abstract

The properties of the state of polarization (SOP) and the degree of polarization (DOP) of Rayleigh backscattered light (RBL) in single-mode fibers (SMF) are investigated theoretically and experimentally when the incident probe is a perfectly coherent continuous-wave (CW) light. It is concluded that the instantaneous DOP of the coherently superposed RBL is always 100%, and the instantaneous SOP is determined by the distributions of the birefringence and the optical phase along the SMF. Therefore, the instantaneous SOP of the coherently superposed RBL does not have a constant relationship with the SOP of the incident CW probe. Furthermore, the instantaneous SOP varies randomly with time because the optical phase is very sensitive to ambient temperature and vibration even in the lab environment. Further theoretical derivation and experimental verification demonstrate, for the first time, that the temporally averaged SOP of the coherently superposed RBL has a simple constant relationship with the SOP of the incident CW probe, and the temporally averaged DOP is 1/3 in an SMF with low and randomly distributed birefringence. The derived formulas and obtained findings can be used to enhance the modelling and improve the performances of phase-sensitive optical time-domain reflectometry and other Rayleigh backscattering based fiber-optic sensors.

## 1. Introduction

Generated from the inherent Rayleigh scattering process in single-mode fibers (SMF), Rayleigh backscattered light (RBL) can affect or even deteriorate the performances of the fiber-optic communication or sensing systems [1,2]. On the other hand, RBL has also been used as the probe signal in distributed fiber sensing systems, such as phase-sensitive optical time-domain reflectometry (φ-OTDR) and polarimetric optical time-domain reflectometry [3,4]. For decades, the RBL in SMF has been comprehensively studied in terms of its statistical properties, spectral properties, polarization properties, etc. [5,6,7]. The polarization properties of the RBL generated from a single Rayleigh scatterer are determined by a simple constant Jones matrix or Mueller matrix, which is the same as that of a mirror, representing reflection [8]. If a continuous-wave (CW) light is launched into an SMF, the resultant RBL is the superposition of the reflected beams from all Rayleigh scatterers which are intensively and randomly distributed along the SMF. The polarization properties of such a superposed RBL are governed by a more complicated Jones matrix and/or a more complicated Mueller matrix, and dependent on the coherence length of the CW light source.

When the coherence length of the light source is much shorter than the effective fiber length, the Mueller matrix governing the polarization properties of the incoherently superposed RBL has been derived by M. Oskar van Deventer [7]. With a CW probe launching into an SMF having low and randomly distributed birefringence, he found that the state of polarization (SOP) of the incoherently superposed RBL is the same as the SOP of the incident CW light, and the degree of polarization (DOP) of the incoherently superposed RBL in one-third of the DOP of the incident CW probe [7].

When the coherence length of the CW light source is longer than the effective fiber length, Tuanwei Xu et al. deduced the corresponding Jones matrix and used the CW probe to study the polarization characteristics of the coherently superposed RBL. Their results show that the SOP of the coherently superposed RBL is the same as the incident CW probe and is not affected by the perturbation applied to the SMF [9]. However, such a perturbation-resistant property is not consistent with the fact that the signal of φ-OTDR is very sensitive to the perturbations applied to the SMF [10,11,12]. It will be demonstrated in this paper that the above conclusions about the SOP of the coherently superposed RBL are incorrect.

The most popular research topic for pulsed incident light is φ-OTDR. φ-OTDR is a promising distributed acoustic sensing technique that utilizes coherently superposed RBL in SMF as a sensing signal [13,14,15,16,17,18]. The polarization properties of coherently superposed RBL play an important role in φ-OTDR. Yixin Zhang et al. found that local birefringence changes in SMF lead to the generation of polarization-dependent noise and the failure to identify multipoint vibration events [19]. Sterenn Guerrier et al. presented a simple and incomplete calculation of the round-trip Jones matrix in φ-OTDR [20,21]. However, a complete analysis of the polarization properties of the coherently superposed RBL has not been reported, to the best of our knowledge. As a direct result, the polarization effects are not even considered in the reported physical models of φ-OTDR [22,23]. In contrast to the perfectly coherent CW probe, the φ-OTDR pulsed light has a finite linewidth. Moreover, there is even chirped-pulse φ-OTDR [24]. Hence, strictly speaking, the φ-OTDR signal is a partially coherently superposed RBL with correlated initial optical phases between different wavelength components. The polarization properties of the φ-OTDR signal should be similar to the polarization properties of the coherently superposed RBL, but more complicated.

In this paper, as the first step towards fully understand the polarization properties of the φ-OTDR signal we aim to derive the Jones matrix and the Mueller matrix governing the coherently superposed RBL and investigate the properties of the SOP and the DOP when the incident light is a perfectly coherent CW probe. In Section 2, two commonly used reference systems and the corresponding round-trip Jones matrices and Muller matrices are summarized as the starting point for theoretical derivation. In Section 3, the Jones matrix and the Mueller matrix governing the coherently superposed RBL are derived. In Section 4, the depolarization in wavelength and the temporal depolarization are investigated. In Section 5, the experiments are conducted to verify the theoretical predictions obtained in Section 3 and Section 4.

## 2. Reference System, Round-Trip Jones Matrix and Muller Matrix

Two commonly used reference systems for the round-trip polarization analysis were proposed by R. C. Jones and Pistoni, respectively [9,25,26,27]. We summarized the forward, the reflection, the backward, and the round-trip Jones matrices in two reference systems in Table 1. The function “$diag\left(a,b,\cdots \right)$” denotes the diagonal matrix with the diagonal elements a,b,⋯. The superscript “T” denotes the matrix transpose. Commenting on or comparing the two different reference systems is beyond the scope of this paper.

We have also derived and summarized the corresponding Mueller matrices in two reference systems in Table 2.

Note that the matrix $R=diag\left(1,1,1,-1\right)$ does not represent the reflection as stated in [7]. It results from the conversion from the Jones matrix $J^{T}\;$ to the corresponding Mueller matrix ${RM}^{T}R$. Obviously, in Jones’ and Pistoni’s reference systems, the difference between two round-trip Jones matrices is the constant matrix A, and the difference between two round-trip Mueller matrices is the constant matrix $A_{M}$. Note that we will use the Jones’ reference system for the theoretical calculations in this paper. If Pistoni’s reference system is preferred, the calculation can be amended by left multiplying A or $A_{M}$. The selection of the reference system does not affect the main conclusions of this paper about the SOP and the DOP of the coherently superposed RBL.

Because an SMF is purely birefringent, the forward Jones matrix J should be unitary, which means that
(1)$$J^{+}J=e^{-\rho }I_{2}$$
where the superscript “+” represents the conjugate transpose of a matrix, and the parameter “ρ” represents the attenuation coefficient of the SMF. Based on Equation (1), R. C. Jones expressed J as [28]
(2)$$J=e^{-\frac{\rho }{2}+i\varphi }\left[\begin{array}{cc} u_{1} & u_{2}\\ -u_{2}^{*} & u_{1}^{*} \end{array}\right]$$
where $i=\sqrt{-1}$, the superscript “*” represents the conjugate of a complex number, and the parameter “φ” is the common optical phase. Note that $u_{1}u_{1}^{*}+u_{2}u_{2}^{*}=1$ so that Equation (2) satisfies Equation (1).

The forward Mueller matrix M, corresponding to the forward Jones matrix J, is an orthogonal matrix. For convenience, it can be written in a general form that was used in [7]
(3)$$M=e^{-\rho }\left[\begin{array}{cccc} 1 & 0 & 0 & 0\\ 0 & m_{1} & m_{2} & m_{3}\\ 0 & m_{4} & m_{5} & m_{6}\\ 0 & m_{7} & m_{8} & m_{9} \end{array}\right]$$

The nine matrix elements are not independent because $M^{T}M=e^{-2\rho }I_{4}$, and they can be expressed as Equation (15) in [29]. Using the formulas connecting the Jones matrix elements and the Mueller matrix elements [30], we have
(4)$$\left\{\begin{array}{c} m_{7}=i\left(u_{1}^{*}u_{2}^{*}-u_{1}u_{2}\right)\;\;\;\;\;\;\;\;\;\;\;\;\;\;\;\;\;\;\;\\ m_{8}=i\left(u_{1}^{2}-u_{2}^{2}-u_{1}^{*2}+u_{2}^{*2}\right)/2\\ m_{9}=\left(u_{1}^{2}+u_{2}^{2}+u_{1}^{*2}+u_{2}^{*2}\right)/2\;\; \end{array}\right.$$

In an SMF, when an incident light propagates forward, it is reflected by a single Rayleigh scatterer with an intensity reflection coefficient γ, and then the resultant RBL propagates backwards to the input end. Hence, the round-trip Jones matrix, in Jones’ reference system, is
(5)$$J_{RT}=\sqrt{\gamma }J^{T}J=\sqrt{\gamma }e^{-\rho +2i\varphi }\left[\begin{array}{cc} u_{1}^{2}+u_{2}^{*2} & u_{1}u_{2}-u_{1}^{*}u_{2}^{*}\\ u_{1}u_{2}-u_{1}^{*}u_{2}^{*} & u_{1}^{*2}+u_{2}^{2} \end{array}\right]\;\;\;\;=\sqrt{\gamma }e^{-\rho +2i\varphi }\left[\begin{array}{cc} m_{9}-im_{8} & im_{7}\\ im_{7} & m_{9}+im_{8} \end{array}\right]$$

Note that Equation (4) has been used to obtain Equation (5). It is easy to verify that $J_{RT}/\sqrt{\gamma }$ satisfies Equations (1) and (2), which means that $J_{RT}$ is still representing pure birefringence. In addition, note that $\sqrt{m_{7}^{2}+m_{8}^{2}+m_{9}^{2}}=1$, which means that $\hat{m}=\left[m_{7},\;m_{8},\;m_{9}\right]$ is a unit vector [7].

## 3. Jones Matrix and Mueller Matrix Governing Coherently Superposed RBL

It is assumed that there are in total N Rayleigh scatterers in the SMF under investigation. When a perfectly coherent CW probe, with the input SOP ${\vec{E}}_{in}$, is launched into the SMF, the SOP of the outgoing coherently superposed RBL at the input end should be
(6)$${\vec{E}}_{out}=\left(\sum_{k=1}^{N}J_{RTk}\right){\vec{E}}_{in}=J_{RTS}{\vec{E}}_{in}$$

The Jones matrix $J_{RTS}$ is the sum of N individual round-trip Jones matrices $J_{RTk},k=1,2,\ldots ,N$, which governs the polarization properties of the outgoing coherently superposed RBL. Next, some symbols are defined as below
(7)$$\left\{\begin{array}{c} p_{j}=\sum_{k=1}^{N}c_{k}m_{jk}\cos2\varphi_{k}\;\;\;\;\;\;\;\;\;\;\;\;\;\;\;\\ q_{j}=\sum_{k=1}^{N}c_{k}m_{jk}\sin2\varphi_{k}\;\;\;\;\;\;\;\;\;\;\;\;\;\;\;\;\\ \vec{p}=\left[p_{7}\;p_{8}\;p_{9}\right],\;\;\vec{q}=\left[q_{7}\;q_{8}\;q_{9}\;\right]\;\; \end{array}\right.j=7,8,9$$
where $c_{k}=\sqrt{\gamma_{k}}e^{-\rho_{k}}$. Then by using Equations (5) and (6), we have
(8)$$J_{RTS}=\left[\begin{array}{cc} t_{1} & t_{2}\\ t_{2} & t_{3} \end{array}\right]$$

In Equation (8), the matrix elements are
(9)$$\left\{\begin{array}{c} t_{1}=q_{8}+p_{9}+i\left(q_{9}-p_{8}\right)\;\;\;\\ t_{2}=-q_{7}+ip_{7}\;\;\;\;\;\;\;\;\;\;\;\;\;\;\;\;\;\;\;\;\;\;\;\\ t_{3}=-q_{8}+p_{9}+i\left(q_{9}+p_{8}\right) \end{array}\right.$$

The Jones matrix $J_{RTS}$ governs the polarization properties of the coherently superposed RBL in the SMF, which represents both birefringence and polarization-dependent loss. Because every element of $J_{RTS}$ is the function of the forward Mueller matrix elements $m_{7},m_{8},m_{9}$ and the optical phase φ, the polarization properties of the coherently superposed RBL depend on the actual birefringence and optical phase distributions along the SMF. There is no simple constant relationship, such as the claimed relationship in [9], between the input SOP of the CW probe and the instantaneous output SOP of the coherently superposed RBL.

Using the formulas between the Jones matrix elements and the Mueller matrix elements [30], the corresponding Mueller matrix $M_{RTS}$ can be calculated as
(10)$$M_{RTS}=\left[\begin{array}{rrrr} w_{0} & w_{1} & w_{2} & w_{3}\\ w_{1} & v_{11} & v_{12} & v_{13}\\ w_{2} & v_{12} & v_{22} & v_{23}\\ {-w}_{3} & {-v}_{13} & {-v}_{23} & {-v}_{33} \end{array}\right]$$

The matrix elements are
(11){w0=p→2+q→2=p72+q72+p82+q82+p92+q92=∑k=1Nck2+∑k,l=1;k≠lNckclm^k·m^lcos⁡2Δφklw1=2q8p9−p8q9=∑k,l=1;k≠lNckclm8km9lsin⁡2Δφklw2=2p7q9−q7p9=∑k,l=1;k≠lNckclm9km7lsin⁡2Δφklw3=2q7p8−p7q8=∑k,l=1;k≠lNckclm7km8lsin⁡2Δφklv11=−p72−q72+p82+q82+p92+q92=∑k=1Nck21−2m7k2+∑k,l=1;k≠lNckclm^k·m^l−2m7km7lcos⁡2Δφklv22=p72+q72−p82−q82+p92+q92=∑k=1Nck21−2m8k2+∑k,l=1;k≠lNckclm^k·m^l−2m8km8lcos⁡2Δφklv33=p72+q72+p82+q82−p92−q92=∑k=1Nck21−2m9k2+∑k,l=1;k≠lNckclm^k·m^l−2m9km9lcos⁡2Δφklv12=−2p7p8+q7q8=−2∑k=1Nck2m7km8k−2∑k,l=1;k≠lNckclm7km8lcos⁡2Δφklv13=−2p7p9+q7q9=−2∑k=1Nck2m7km9k−2∑k,l=1;k≠lNckclm7km9lcos⁡2Δφklv23=−2p8p9+q8q9=−2∑k=1Nck2m8km9k−2∑k,l=1;k≠lNckclm8km9lcos⁡2Δφkl
where $∆\varphi_{kl}=\varphi_{k}-\varphi_{l}$.

It is verified that $M_{RTS}^{T}GM_{RTS}=\sqrt{\left|M_{RTS}\right|}G$. Here, $\left|M_{RTS}\right|$ is the determinant of $M_{RTS}$, and $G=diag\left(1,-1,-1,-1\right)$. Therefore, $M_{RTS}$ describes a non-depolarizing optical system [31]. It means that the instantaneous DOP of the coherently superposed RBL is always 100% when the DOP of the incident coherent probe is 100%.

## 4. Depolarization

If the light source has a finite linewidth, the Jones matrix and the Mueller matrix governing the RBL induced by each wavelength component within the linewidth range can still be described by Equations (8) and (10), respectively. However, the overall Jones matrix does not exist because the Jones matrices for different wavelength components cannot be added incoherently. The overall Mueller matrix should be the sum of all wavelength-resolved sub-Mueller matrices depicted in Equation (10) because of the incoherent addition of lights with different wavelengths and uncorrelated optical phases. Furthermore, we assume the following conditions can be ensured that (1) $∆\varphi_{kl}$ and ${\hat{m}}_{k}$ vary independently with respect to the optical wavelength, (2) $c_{k}$ can be considered to be wavelength-independent within the linewidth range, (3) $∆\varphi_{kl}$ varies over a wide range (≫2π) with respect to the optical wavelength within the linewidth range. Then it has
(12)$${\langle sin2\Delta \varphi_{kl}\rangle }_{\lambda }=0,\;\;{\langle cos2\Delta \varphi_{kl}\rangle }_{\lambda }=\left\{\begin{array}{c} 1\;\;\;\;\;k=l\\ 0\;\;\;\;\;k\neq l \end{array}\right.$$
where “${\langle \cdot \rangle }_{\lambda }$” denotes the average operation over the optical wavelength. Then based on Equations (11) and (12), and using ${\hat{m}}_{k}\cdot {\hat{m}}_{k}=1$, the wavelength-averaged Mueller matrix is
(13)$${\langle M_{RTS}\rangle }_{\lambda }=\sum_{k=1}^{N}c_{k}^{2}\left[\begin{array}{cccc} 1 & 0 & 0 & 0\\ 0 & 1-2{\langle m_{7k}^{2}\rangle }_{\lambda } & -2{\langle m_{7k}m_{8k}\rangle }_{\lambda } & -2{\langle m_{7k}m_{9k}\rangle }_{\lambda }\\ 0 & -2{\langle m_{7k}m_{8k}\rangle }_{\lambda } & 1-2{\langle m_{8k}^{2}\rangle }_{\lambda } & -2{\langle m_{8k}m_{9k}\rangle }_{\lambda }\\ 0 & 2{\langle m_{7k}m_{9k}\rangle }_{\lambda } & 2{\langle m_{8k}m_{9k}\rangle }_{\lambda } & 2{\langle m_{9k}^{2}\rangle }_{\lambda }-1 \end{array}\right]\;=\langle \sum_{k=1}^{N}c_{k}^{2}\left[\begin{array}{cccc} 1 & 0 & 0 & 0\\ 0 & 1-2m_{7k}^{2} & -2m_{7k}m_{8k} & -2m_{7k}m_{9k}\\ 0 & -2m_{7k}m_{8k} & 1-2m_{8k}^{2} & -2m_{8k}m_{9k}\\ 0 & 2m_{7k}m_{9k} & 2m_{8k}m_{9k} & 2m_{9k}^{2}-1 \end{array}\right]\rangle_{\lambda }$$

Comparing this with the results obtained in [7], it can be found that Equation (13) is the same as the Mueller matrix governing incoherently superposed RBL when the coherence length of the laser source is much shorter than the effective fiber length.

When the SMF under investigation is randomly birefringent and the effective fiber length is far longer than the polarization beat length, it was experimentally demonstrated, using broadband light source, that [7]
(14)$$\langle \sum_{k=1}^{N}c_{k}^{2}\left[\begin{array}{cccc} 1 & 0 & 0 & 0\\ 0 & 1-2m_{7k}^{2} & -2m_{7k}m_{8k} & -2m_{7k}m_{9k}\\ 0 & -2m_{7k}m_{8k} & 1-2m_{8k}^{2} & -2m_{8k}m_{9k}\\ 0 & 2m_{7k}m_{9k} & 2m_{8k}m_{9k} & 2m_{9k}^{2}-1 \end{array}\right]\rangle_{\lambda }\propto \frac{1}{3}\left[\begin{array}{cccc} 1 & 0 & 0 & 0\\ 0 & 1 & 0 & 0\\ 0 & 0 & 1 & 0\\ 0 & 0 & 0 & -1 \end{array}\right]=\frac{1}{3}R$$ where “∝” means “proportional to”. In this case, it has ${\langle M_{RTS}\rangle }_{\lambda }\propto R/3$. This means when the incident light is completely polarized, the DOP of the outgoing incoherently superposed RBL is 1/3 [7]. The consistency with the reported result in [7] partially confirms the correctness of Equations (8) and (10).

The optical phase in an SMF is very sensitive to the variation of the ambient temperature and the vibration applied to the SMF. Therefore, the SOP of the coherently superposed RBL varies rapidly with time even in the lab environment. At a given time, the instantaneous SOP can be anywhere on the Poincaré sphere. At the same time, the instantaneous DOP should remain at 100%. If the optical phase varies randomly over a long period of time so that $∆\varphi_{kl}$ varies over a wide range (≫2π), the following equation is valid
(15)$${\langle sin2\Delta \varphi_{kl}\rangle }_{t}=0,\;\;\;\;\;\;{\langle cos2\Delta \varphi_{kl}\rangle }_{t}=\left\{\begin{array}{c} 1\;\;\;\;\;k=l\\ 0\;\;\;\;\;k\neq l \end{array}\right.$$ where “${\langle \cdot \rangle }_{t}$” represents the average operation over time. Furthermore, $∆\varphi_{kl}$ and ${\hat{m}}_{k}$ can be considered to vary independently with time. Then
(16)$${\langle M_{RTS}\rangle }_{t}=\sum_{k=1}^{N}c_{k}^{2}\left[\begin{array}{cccc} 1 & 0 & 0 & 0\\ 0 & 1-2{\langle m_{7k}^{2}\rangle }_{t} & -2{\langle m_{7k}m_{8k}\rangle }_{t} & -2{\langle m_{7k}m_{9k}\rangle }_{t}\\ 0 & -2{\langle m_{7k}m_{8k}\rangle }_{t} & 1-2{\langle m_{8k}^{2}\rangle }_{t} & -2{\langle m_{8k}m_{9k}\rangle }_{t}\\ 0 & 2{\langle m_{7k}m_{9k}\rangle }_{t} & 2{\langle m_{8k}m_{9k}\rangle }_{t} & 2{\langle m_{9k}^{2}\rangle }_{t}-1 \end{array}\right]\;\;\;\;=\langle \sum_{k=1}^{N}c_{k}^{2}\left[\begin{array}{cccc} 1 & 0 & 0 & 0\\ 0 & 1-2m_{7k}^{2} & -2m_{7k}m_{8k} & -2m_{7k}m_{9k}\\ 0 & -2m_{7k}m_{8k} & 1-2m_{8k}^{2} & 2m_{8k}m_{9k}\\ 0 & 2m_{7k}m_{9k} & 2m_{8k}m_{9k} & 2m_{9k}^{2}-1 \end{array}\right]\rangle_{t}$$

Equation (16) implies that the time-averaged $M_{RTS}$ is also a Mueller matrix with a depolarization effect. If the SMF is randomly birefringent and the effective fiber length is much longer than the polarization beat length, it also has a proportionality equation
(17)$${\langle M_{RTS}\rangle }_{t}\propto \frac{1}{3}R$$

In this case, the time-averaged output SOP of the coherently superposed RBL has a constant relationship with the input SOP, and the time-averaged output DOP is 1/3.

## 5. Experiments

In this section, we will experimentally verify two theoretical predictions mentioned above. The first experiment is to verify (1) the instantaneous SOP of the coherently superposed RBL using a CW probe varies randomly with time and has no constant relationship with the input SOP, (2) the instantaneous DOP is always 100%. The second experiment is to verify that the time-averaged DOP of the output coherently superposed RBL is 1/3 in an SMF with low and randomly distributed birefringence.

### 5.1. Instantaneous SOP and DOP Measurement

The experimental setup is illustrated in Figure 1. In this experiment, the fiber under test is a 24 km long SMF with low and randomly distributed birefringence. The 1550 nm narrow linewidth laser used in this experiment has a linewidth of less than 100 Hz (coherence length longer than 2000 km) and a CW output power of 5 mW. The polarimeter has a sampling rate up to 4 MHz, which means that a SOP measurement can be done within 250 ns.

Due to the long effective fiber length and the narrow laser linewidth in this experiment, the power of the stimulated Brillouin scattering (SBS) light exceeds the power of the RBL. To suppress the SBS light, a tunable optical filter is used. The optical spectrum in Figure 2 shows that the RBL dominates by more than 30 dB after passing through the tunable optical filter. Due to the large insertion loss of the tunable optical filter, using an erbium-doped fiber amplifier (EDFA) to increase the power of the RBL enables the polarimeter to accurately measure the SOP and the DOP.

Figure 3 shows 1000 successive but independent measurement results of the optical power, the normalized Stokes parameters, and the DOP over 250 µs. Each measurement takes 250 ns. Clearly, the instantaneous DOP remains at 100%. Meanwhile, even in a laboratory environment, the optical power and the SOP change rapidly and randomly due to the environmental disturbances.

### 5.2. Temporal Depolarization Measurement

When the time of each measurement using the polarimeter is increased to 1 ms, 1000 successive but independent measurements of the DOP are performed. The DOP measurement results in Figure 4 display the temporal depolarization effect induced by the ambient disturbances. The DOP can be any value between 0 and 1, depending on the actual environmental disturbances applied to the SMF.

The longest time for each measurement using the polarimeter is 10 s. Then 300 independent measurements are performed with the individual measurement time of 10 s. The *k*-th averaged DOP with an average time of 10×k seconds is calculated by
(18)$${DOP}_{k}=\frac{\sqrt{{\left(\sum_{j=1}^{k}s_{1j}\right)}^{2}+{\left(\sum_{j=1}^{k}s_{2j}\right)}^{2}+{\left(\sum_{j=1}^{k}s_{3j}\right)}^{2}}}{\sum_{j=1}^{k}s_{0j}}$$
where $\left[s_{0j},\;s_{1j},\;s_{2j},\;s_{3j}\right]$ are the four Stokes parameters that are measured in the j-th measurement. Through such calculations, the evolution of the DOP is shown in Figure 5. The DOP converges to 1/3 when the accumulated measurement time is 2500 s. This result confirms the validity of the theoretical prediction in Equation (17).

## 6. Conclusions

The Jones matrix and the Mueller matrix governing the coherently superposed RBL in SMF with a CW probe are derived and verified via experiments for the first time. Based on these polarization matrices, the instantaneous SOP of the coherently superposed RBL depends on the actual distributions of the birefringence and the optical phase along the SMF, which has no constant relationship with the input SOP of the CW probe. Meanwhile, the instantaneous DOP remains at 100%. It is also predicted that the summation of the wavelength-resolved or the time-resolved Mueller matrices governing the coherently superposed RBL leads to the Mueller matrix governing the incoherently superposed RBL. Then the time-averaged DOP of the coherently superposed RBL will converge to 1/3 when the SMF is randomly birefringent and much longer than the polarization beat length; the time-averaged SOP has a constant relationship with the input SOP. Two experiments are performed to validate the theoretical predictions about the instantaneous SOP and DOP, and the time-averaged DOP. Good agreements between the theoretical predictions and the experimental results are evident.

In addition, it should be highlighted that when the incident probe is pulsed light as in a φ-OTDR system, the Jones matrix and the Mueller matrix derived in this paper can be used to qualitatively study the polarization properties of the φ-OTDR signal. Strictly speaking, the φ-OTDR signal is a partially coherently superposed RBL with correlated initial optical phases among different wavelength components. Therefore, a rigorous analysis of the φ-OTDR signal could start from the formulas obtained in this paper.

## Figures and Tables

**Figure 1 sensors-23-07769-f001:**
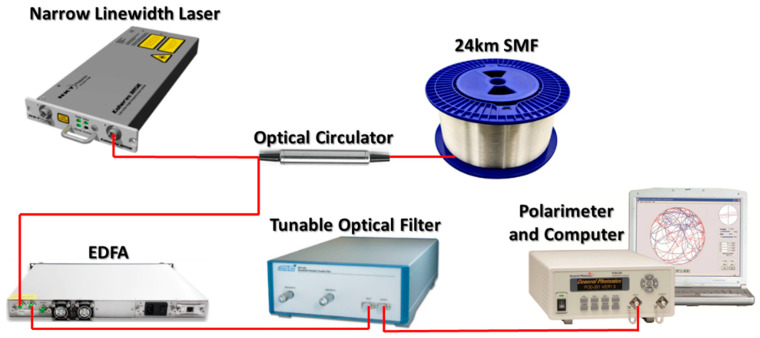
The experimental setup to measure the instantaneous SOP and DOP of the coherently superposed RBL in a 24 km long SMF. EDFA: erbium-doped fiber amplifier.

**Figure 2 sensors-23-07769-f002:**
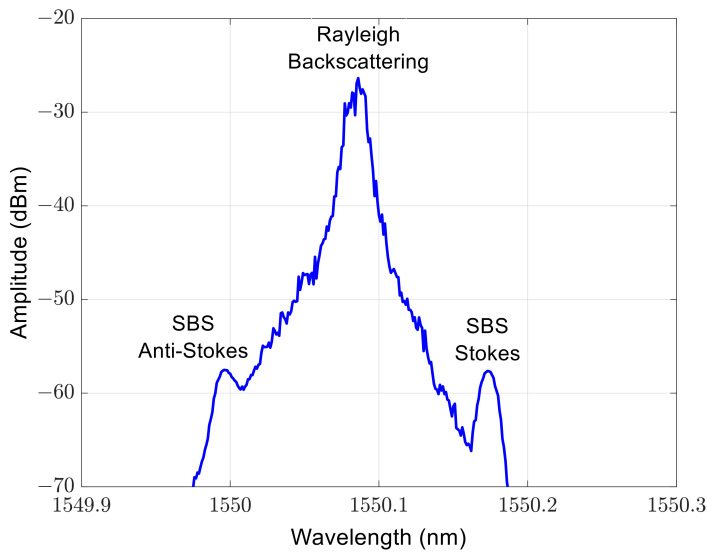
The optical spectrum of the light after passing through the tunable optical filter. The SBS light has been effectively suppressed and the RBL dominates by more than 30 dB.

**Figure 3 sensors-23-07769-f003:**
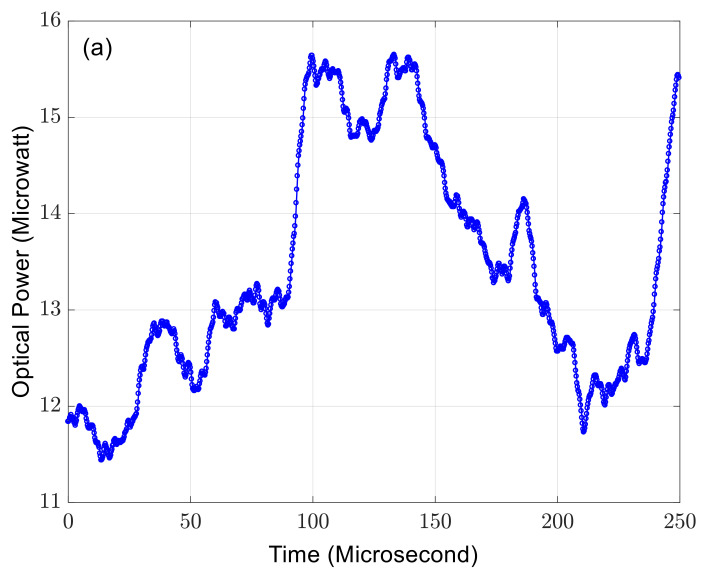
The 1000 independent measurement results of the instantaneous (**a**) optical power, (**b**) normalized Stokes parameters, and (**c**) DOP. One measurement takes 250 ns.

**Figure 4 sensors-23-07769-f004:**
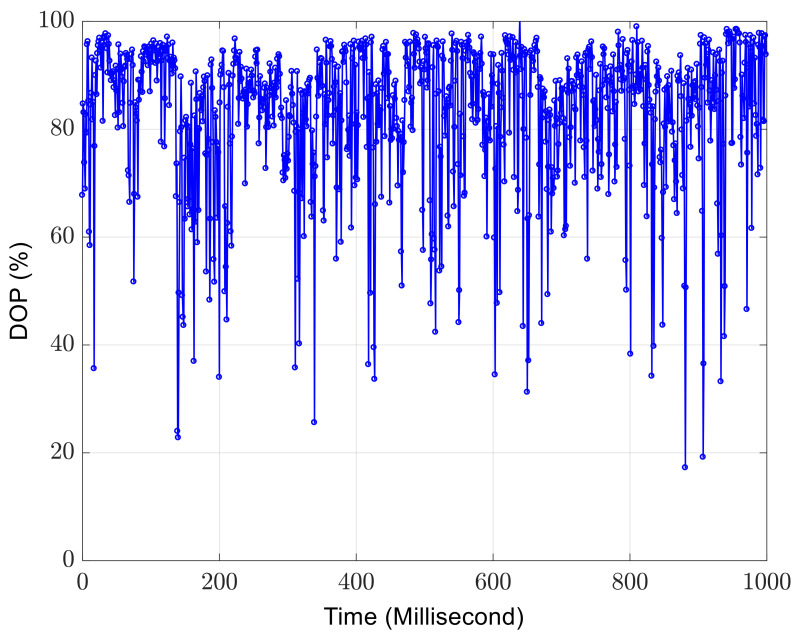
The 1000 measurement results of DOP. One measurement takes 1 ms. In this case, temporal depolarization occurs.

**Figure 5 sensors-23-07769-f005:**
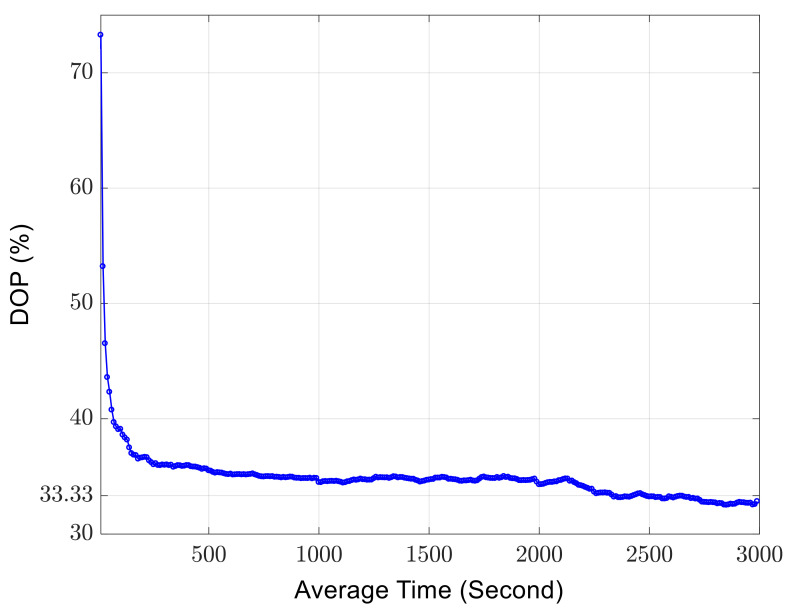
The DOP evolution with respect to the accumulated measurement time. With a sufficient long measurement time, the DOP converges to 1/3.

**Table 1 sensors-23-07769-t001:** Jones matrices in Jones’ and Pistoni’s reference systems.

	Jones’ Reference System	Pistoni’s Reference System
Forward	J	J
Reflection	$I_{2}=diag\left(1,1\right)$	$A=diag\left(1,-1\right)$
Backward	$J^{T}$	$AJ^{T}A$
Round-Trip	$J^{T}J$	$AJ^{T}J$

**Table 2 sensors-23-07769-t002:** Mueller matrices in Jones’ and Pistoni’s reference systems.

	Jones’ Reference System	Pistoni’s Reference System
Forward	M	M
Reflection	$I_{4}=diag\left(1,1,1,1\right)$	$A_{M}=diag\left(1,1,-1,-1\right)$
Backward	${RM}^{T}R$	${A_{M}RM}^{T}RA_{M}$
Round-Trip	${RM}^{T}RM$	$A_{M}{RM}^{T}RM$

## Data Availability

The data in this work are available from the corresponding author upon request.

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
