# Peer review of "Polarization Properties of Coherently Superposed Rayleigh Backscattered Light in Single-Mode Fibers"

_sensors, 2023, doi:10.3390/s23187769_

Round 1

Reviewer 1 Report

The authors should address the following:

- define how you calculate J 

- check the numerator in Eq. (14a)

- figures are cramped - try to increase their size

- clarify the use of polarisation controllers

- explain the use of the pixelated polarizers in the method

- explain the limitations of the method, particularly the wavelength range of a polarising camera sensor 

There are a few typos in the manuscript. Overall, the good quality of the  English language is throughout the paper.

Reviewer 2 Report

The paper presents the experimental study of the polarization properties  of the  backscattered light in an optical fiber for the conditions of light  coherence length bigger than the   fiber length. It is shown that the instantaneous polarization state is well defined, but it can be anywhere on  Poincare sphere, independently on the source polarization. The variations of polarization properties on millisecond and second time scale due to random variations of fiber properties reduce the degree of polarization at these  times. 

     The results can be useful for  fiber sensor implementation.  

     The paper is well written. 

      I consider, that it can be published as is.

Author Response

Thanks for your comments.

Reviewer 3 Report

In this paper, the authors derived the Jones matrix and the Muller matrix of coherent Rayleigh backscattering in single-mode fibres and predicted the properties of the SOP and the DOP of the Rayleigh backscattered light accordingly. Moreover, they conducted the experiments to validate some theoretical predictions successfully. Based on my research background, I think these fundamental new knowledges are of great importance for the R&D of Rayleigh backscattering based distributed fiber sensors, especially phase-sensitive OTDR. The obtained results in this paper may attract interests and attentions in the relevant research communities. I suggest that this paper should be accepted by Sensors.

Meanwhile, I have some minor revision suggestions and questions as follows:

1)      Page 2, paragraph 2. The sentence “However, such a perturbation-resistant property is not consistent with the signal features…….”. What signal features do the authors refer to?

2)      Page 2, paragraph 3. In this paragraph, the authors used the word “superimposed”. But in other sections, including the title, the authors used the word “superposed”. Why? If they have the same meaning, please just use the same word in the entire paper.

3)      The format of the equations in Section 3 seems different from that of other equations. Please use the same format in the entire paper.

4)      The authors mentioned that JRT in Equation (5) represents pure birefringence. Then what polarization effects does JRTS in Equation (8) represent?

5)      The authors claimed that the DOP of the time averaged RBL is 1/3. Is this valid for any  SMF and/or under what circumstances?

6)      The authors claimed that the conclusions in reference [9] are incorrect. And in the experiments, it was highlighted that SBS light should be suppressed. Does this mean that the polarization properties described in [9] are for SBS light rather than RBL?

7)      The following two papers are well relevant with the works in this paper. The authors may consider citing them as the references.

[1] Z. Zhao, et al., "Interference fading suppression in φ-OTDR using space-division multiplexed probes," Opt. Express 29, 15452-15462, 2021.

[2] Z. Lin, et al, "Frequency Response Enhanced Quantitative Vibration Detection Using Fading-Free Coherent φ-OTDR With Randomized Sampling," in Journal of Lightwave Technology, vol. 41, no. 15, pp. 5159-5168, 1 Aug.1, 2023.
